# A Simple and Efficient Smoothing Method for Faster Optimization and Local Exploration

**Kevin Scaman**    **Ludovic Dos Santos**    **Merwan Barlier**    **Igor Colin**

Huawei Noah's Ark Lab

## Abstract

This work proposes a novel smoothing method, called *Bend, Mix and Release (BMR)*, that extends two well-known smooth approximations of the convex optimization literature: randomized smoothing and the Moreau envelope. The BMR smoothing method allows to trade-off between the computational simplicity of randomized smoothing (RS) and the approximation efficiency of the Moreau envelope (ME). More specifically, we show that BMR achieves up to a $\sqrt{d}$ multiplicative improvement compared to the approximation error of RS, where $d$ is the dimension of the search space, while being less computation intensive than the ME. For non-convex objectives, BMR also has the desirable property to widen local minima, allowing optimization methods to reach small cracks and crevices of extremely irregular and non-convex functions, while being well-suited to a distributed setting. This novel smoothing method is then used to improve first-order non-smooth optimization (both convex and non-convex) by allowing for a local exploration of the search space. More specifically, our analysis sheds light on the similarities between evolution strategies and BMR, creating a link between exploration strategies of zeroth-order methods and the regularity of first-order optimization problems. Finally, we evidence the impact of BMR through synthetic experiments.

## 1   Introduction

First-order optimization is at the heart of most training algorithms in Machine Learning. Thanks to backpropagation, neural networks have achieved a number of impressive successes over the past few years, including natural language processing [1], image processing [2], and reinforcement learning [3]. However, training complex state of the art architectures remains difficult due to the highly non-smooth and non-convex nature of their loss functions [4]. While stochastic gradient descent and its variants (Adam [5], RMSProp [6], or Nesterov's accelerated gradient descent [7]) showed surprisingly good performance in many practical scenarios, these algorithms remain fragile for particularly non-smooth or non-convex objectives [8].

Smooth approximations, such as Randomized Smoothing (RS) [9] or the Moreau Envelope (ME) [10], provide an elegant solution to deal with both problems at the same time. In order to solve optimization problems of the form $\min_{x \in \mathbb{R}^d} f(x)$ where $f : \mathbb{R}^d \to \mathbb{R}$ is a non-smooth (possibly non-convex) and $L$-Lipschitz objective function, these methods replace the objective function with a smooth approximation of it, which allows for accelerated convergence rates and parameter space exploration. This idea was successfully used in convex [11, 12], distributed [13, 14], composite [15, 16, 17], and non-convex optimization [18] to improve convergence rates and significantly reduce computation times. In distributed optimization, RS, defined by $f_{\mu}^{RS}(x) = \mathbb{E}\left[f(x + \mu X)\right]$ where $X$ is a Gaussian normal random variable, allowed to use extra computations to accelerate convergence [9, 13]. Its simple form has the advantage of being easy to compute and parallelize. Unfortunately, RS suffers from a $\sqrt{d}$ multiplicative factor in its approximation error, leading to a $d^{1/4}$ multiplicative term in the convergence rate of distributed algorithms (e.g. DRS [13] or PPRS [14]), which can be

Table 1: Strengths and weaknesses of smoothing methods for convex $L$-Lipschitz functions. For ME, we use the parameter $\mu' = \mu/L$ to match the smoothness of RS.

| Smoothing method | Smoothness | Approx. error | Gradient comp. |
|---|---|---|---|
| Randomized smoothing | $L/\mu$ | $\mu L \sqrt{d}$ | Sampling (easy) |
| Moreau envelope | $L/\mu$ | $\mu L$ | Opt. problem (hard) |
| $\alpha$-BMR (ours) | $L/\mu$ | $\mu L \left( \frac{1}{2} + \sqrt{\frac{d}{\max\{1,\mu L\alpha\}}} \right)$ | Biased sampling (med.) |

prohibitive for high-dimensional problems. Removing this dependency in the dimension is yet an open problem of the field. For composite optimization, the Moreau Envelope (ME), defined by $f_\mu^{ME}(x) = \min_{y \in \mathbb{R}^d} f(y) + \frac{1}{2\mu}\|y - x\|^2$, and its gradient the *proximal operator* [19, 15, 16, 20, 10], allowed to obtain smooth convergence rates even in the presence of non-smooth objectives of the form $\min_x f(x) + g(x)$, where $f$ is smooth and $g$ is non-smooth but simple (i.e., *prox friendly*). While ME is harder to compute, its theoretical guarantees are better, including a dimension-free approximation error (see Table 1 for a comparison of smoothness and approximation error of RS, ME and our novel smoothing operator for $L$-Lipschitz functions). Finally, in non-convex optimization, a relationship between exploration and smoothing was investigated in a recent line of works [21, 22], and provide an elegant theoretical setting to understand exploration in stochastic gradient descent. Moreover, RS was successfully applied to non-smooth non-convex optimization problems and provided the first convergence rates in this particularly difficult setting [18].

This work introduces a novel smooth approximation, called *Bend, Mix and Release* (BMR), that benefits from the advantages of both techniques by biasing the expectation in RS towards low values of the objective function. BMR thus allows to trade-off between computation complexity and quality of the approximation (see Table 1). As for RS, its gradient can be approximated through sampling, which allows to distribute its computation on parallel workers. Moreover, BMR also improves the exploration of non-convex functions by widening deep local minima, and thus allows the optimization algorithm to reach better local optima for particularly irregular objectives. Finally, we show that such a smooth approximation can be used for distributed and non-convex optimization in order to improve upon RS based algorithms.

This paper is organized as follows. First, we introduce BMR smoothing and highlight its key properties in Section 2, including smoothness, approximation error and limit behavior. We then discuss in Section 3 how to use BMR for optimization, with a special focus on the approximation of the gradient. Finally, in Section 4, we show the benefits of BMR over RS empirically on a synthetic non-smooth and non-convex objectives. The proofs of all theorems are available in the supplementary material.

## 2   Bend, Mix and Release

In this section, we introduce our novel smooth approximation, called *Bend, Mix and Release* (BMR). We first recall useful notation and definitions, and then provide a formal definition of BMR. To enhance intuition, we then provide alternative formulations before stating its main properties.

### 2.1   Notation and standard definitions

We denote as $\mathcal{N}(\mu, \Sigma)$ the multivariate Gaussian probability distribution of mean $\mu \in \mathbb{R}^d$ and (semi-definite positive) covariance matrix $\Sigma \in \mathbb{R}^{d \times d}$. For simplicity, we assume hereafter that all functions are measurable and differentiable. A function $f : \mathbb{R}^d \to \mathbb{R}$ is $L$-Lipschitz if, for all $x \in \mathbb{R}^d$, $\|\nabla f(x)\|_2 \leq L$; is convex if, for all $x, y \in \mathbb{R}^d$, $f(y) - f(x) - \nabla f(x)^\top (y - x) \geq 0$; and is $\beta$-smooth if its gradient is $\beta$-Lipschitz continuous, or equivalently $\forall x, y \in \mathbb{R}^d$,

$$-\frac{\beta}{2}\|y - x\|_2^2 \quad \leq \quad f(y) - f(x) - \nabla f(x)^\top (y - x) \quad \leq \quad \frac{\beta}{2}\|y - x\|_2^2. \tag{1}$$

For non-convex functions, convergence results typically only require the second inequality, as it ensures that the objective function is smaller that an auxiliary quadratic function. We thus relax this property to that of *partial smoothness* defined below.

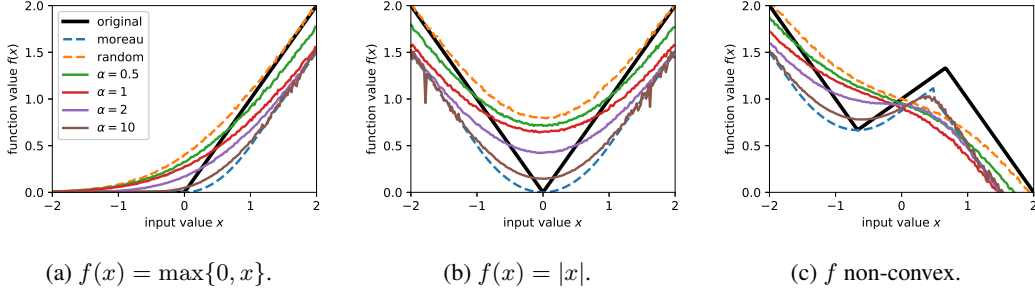

(a) $f(x) = \max\{0, x\}$.  (b) $f(x) = |x|$.  (c) $f$ non-convex.

Figure 1: Effect of the parameter $\alpha$ on BMR smoothing (with $\gamma = \min\{1, \alpha^{-1/2}\}$). When $\alpha \to 0$ (resp. $\alpha \to +\infty$), BMR tends to randomized smoothing (resp. Moreau envelope).

**Definition 1** (Partial smoothness). Let $f : \mathbb{R}^d \to \mathbb{R}$ be a (possibly non-convex) function and $\beta \geq 0$. The function $f$ is $\beta$-upper-smooth (resp. $\beta$-lower-smooth) if, for all $x, y \in \mathbb{R}^d$, $f(y) - f(x) - \nabla f(x)^\top (y - x) \leq \frac{\beta}{2} \|y - x\|_2^2$ (resp. $f(y) - f(x) - \nabla f(x)^\top (y - x) \geq -\frac{\beta}{2} \|y - x\|_2^2$).

**Remark 1.** Convexity is equivalent to 0-lower-smoothness, and if a function is both $\beta$-lower-smooth and $\beta$-upper-smooth, it is then $\beta$-smooth. As a consequence, a convex function that is $\beta$-upper-smooth is also $\beta$-smooth.

## 2.2 BMR smoothing

Despite their differences, RS and ME share a common similarity: both operators are convolutions (in the classical sense for RS, and in tropical geometry for ME). As tropical geometry replaces the sum by a maximum, a natural strategy to interpolate between them consists in biasing the expectation of RS towards the minimum of ME. Following this intuition, our smoothing operator uses a three step procedure (*Bend, Mix and Release*) to achieve this interpolation: first, the function is *bent* to give more importance to low values of the objective function. Then, the function is *mixed* on a Gaussian neighborhood. Finally, the function is *released* using the inverse of the bending function.

**Definition 2** (Bend, Mix and Release). Let $\alpha, \gamma, L > 0$ and $f : \mathbb{R}^d \to \mathbb{R}$ be an $L$-Lipschitz function. The *BMR (Bend, Mix and Release)* smoothing of $f$ is defined as follows:

$$f_{\gamma,\alpha}(x) = \phi_\alpha^{-1} \left( \mathbb{E}\left[ \phi_\alpha\Big( f(x + \gamma X) \Big) \right] \right), \tag{2}$$

where $X \sim \mathcal{N}(0, I)$ is a normal multivariate Gaussian variable and $\phi_\alpha(x) = \frac{1 - e^{-\alpha x}}{\alpha}$.

Alternative formulations of BMR smoothing provide three different insights into the regularization: 1) **Moment generating function:** Using the definition of $\phi_\alpha$, Eq. (2) simplifies to $f_{\gamma,\alpha}(x) = -\alpha^{-1} \ln \mathbb{E}\left[ e^{-\alpha f(x + \gamma X)} \right]$. This quantity can be interpreted as the logarithm of the moment generating function (also known as the cumulant generating function [23]) of $f(x + \gamma X)$, and bears resemblance to the *LogSumExp* (or softmax) operator [24]. Moreover, the simplicity of this notation makes it more convenient for practical computation. 2) **Gaussian filter:** By rewriting the expectation as a convolution with the Gaussian distribution, we get that $f_{\gamma,\alpha}(x) = \phi_\alpha^{-1} \circ \big( (\phi_\alpha \circ f) * p_\gamma \big)(x)$, where $*$ is the convolution and $p_\gamma$ is the Gaussian probability distribution of standard deviation $\gamma$. This notation shows that BMR smoothes by blurring the function $\phi_\alpha \circ f$ using a Gaussian filter, and will be useful for the computation of derivatives. 3) $L_p$**-norm:** Finally, using the notation $\|X\|_p = \mathbb{E}\left[|X|^p\right]^{1/p}$ for any $p > 0$, we have $f_{\gamma,\alpha}(x) = -\ln \left\| e^{-f(x + \gamma X)} \right\|_\alpha$. This notation shows that, as $\alpha$ increases, the logarithm of the smoothing operator tends to the maximum of the function $e^{-f}$. By choosing wisely $\gamma$ as $\alpha$ tends to $+\infty$, we will show in Proposition 2 that the expectation in randomized smoothing tends to the minimum in the Moreau envelope, allowing a continuous transition between the two smoothing operators.

While other bending functions (e.g. power laws) could, in principle, be used, $\phi_\alpha$ has the crucial advantage of being *translation equivariant*. This property increases the stability of the operator, and is fundamental to prove the approximation bounds of Proposition 4.

**Proposition 1.** *Let $\alpha, \gamma \geq 0$, and $f : \mathbb{R}^d \to \mathbb{R}$ a function. The following properties hold:*

1. Let $\tau_a(x) = x + a$ be a translation by a constant $a \in \mathbb{R}^d$. Then $(\tau_a \circ f)_{\gamma,\alpha} = \tau_a \circ f_{\gamma,\alpha}$ and $(f \circ \tau_a)_{\gamma,\alpha} = f_{\gamma,\alpha} \circ \tau_a$.

2. Let $\sigma_c(x) = cx$ be a scaling function by a constant $c \in \mathbb{R}$. Then $(\sigma_c \circ f)_{\gamma,\alpha} = \sigma_c \circ f_{\gamma,c\alpha}$ and $(f \circ \sigma_c)_{\gamma,\alpha} = f_{c\gamma,\alpha} \circ \sigma_c$.

Finally, with the correct choice of parameters, BMR interpolates between RS and ME. More specifically, in order to recover ME, the size of noise $\gamma$ should decrease as the bending factor $\alpha$ increases.

**Proposition 2** (Limit behavior). *Let $f : \mathbb{R}^d \to \mathbb{R}$ be a function. For any $\mu \geq 0$ and $x \in \mathbb{R}^d$,* $\lim_{\alpha \to 0} f_{\mu,\alpha}(x) = f_\mu^{RS}(x)$ *and* $\lim_{\alpha \to +\infty} f_{\sqrt{\frac{\mu}{\alpha}},\alpha}(x) = f_\mu^{ME}(x)$.

## 2.3 Smoothness and approximation error

As discussed in Section 1, smoothness and approximation error are the two key properties that are necessary to use the operator for optimization. Fortunately, the theoretical analysis of BMR provides a bounds for both quantities.

**Proposition 3.** *If $f$ is L-Lipschitz, its BMR smoothing $f_{\gamma,\alpha}$ is* $\min\left\{\frac{L}{\gamma}, \frac{1}{\alpha\gamma^2}\right\}$*-upper-smooth.*

Note that, contrary to RS, the upper-smoothness of BMR is bounded by $\frac{1}{\alpha\gamma^2}$ even for arbitrarily large $L$. Moreover, if $f$ is convex, then $f_{\gamma,\alpha}$ is $\min\left\{L/\gamma, 1/\alpha\gamma^2\right\}$-smooth and smooth optimization algorithms can be used to minimize it. Finally, the approximation error is bounded as follows.

**Proposition 4.** *If $f$ is L-Lipschitz, $f_{\gamma,\alpha}$ is an approximation of $f$ in the following sense: $\forall x \in \mathbb{R}^d$,*

$$|f_{\gamma,\alpha}(x) - f(x)| \leq \frac{\alpha\gamma^2 L^2}{2} + \gamma L \sqrt{d}. \tag{3}$$

With the choice of parameter $\gamma = \min\left\{\mu, \sqrt{\frac{\mu}{L\alpha}}\right\}$, we have that $f_{\gamma,\alpha}$ is $\frac{L}{\mu}$-upper-smooth and

$$|f_{\gamma,\alpha}(x) - f(x)| \leq \mu L \left(\frac{1}{2} + \sqrt{\frac{d}{\max\{1, \mu L\alpha\}}}\right). \tag{4}$$

As discussed in Section 1, this approximation error interpolates between that of RS (i.e., $\mu L\sqrt{d}$) and ME (i.e., $\mu L$), and thus allows to trade-off computation complexity with approximation error. When $\mu L\alpha < 1$, the approximation is in $\mu L\sqrt{d}$ and BMR is similar to randomized smoothing. When $\mu L\alpha > d$, BMR approaches the Moreau envelope and benefits from a $\mu L$ approximation error.

## 2.4 Widening of local minima

Finally, the Moreau envelope has the notable advantage of having no approximation error at the minimum of the function. The same behavior can be observed for BMR as $\alpha$ tends to $+\infty$. Effects of the parameter $\alpha$ on the BMR smoothing is illustrated in Figure 1.

**Proposition 5** (Approximation at the minimum). *If $f$ is L-Lipschitz and $x^* \in \min_{x \in \mathbb{R}^d} f(x)$, then*

$$0 \leq f_{\gamma,\alpha}(x^*) - f(x^*) \leq \frac{d}{2\alpha}\left(1 + \ln\left(1 + \frac{(\alpha\gamma L)^2}{d}\right)\right). \tag{5}$$

For non-convex optimization, this result, combined with the smoothness of $f_{\gamma,\alpha}$, implies the widening of all local minima, allowing the optimization algorithm to find cracks and crevices of the objective function (see Section 4 for an experimental validation).

**Proposition 6** (Widening of local minima). *Let $\mu > 0$ and $y \in \mathbb{R}^d$. If $f$ is L-Lipschitz, then there exists $z \in \mathbb{R}^d$ such that $\|y - z\| \leq \mu$ and, $\forall x \in \mathbb{R}^d$,*

$$f_{\gamma,\alpha}(x) \leq f(y) + \frac{L}{2\mu}\|x - z\|^2 + \frac{d}{2\alpha}\left(1 + \ln\left(1 + \frac{\alpha\mu L}{d}\right)\right), \tag{6}$$

*where $\gamma = \min\left\{\mu, \sqrt{\frac{\mu}{L\alpha}}\right\}$.*

In other words, for every local minima $y \in \mathbb{R}^d$ of the objective function $f$, $f_{\gamma,\alpha}(x)$ will be small (i.e., approximately $f(y)$) in a neighborhood of $y$ of size $\mu$ (see Figure 1c for an example). As a consequence, a good but thin local minimum will have its basin of attraction increased, and thus be easier to reach by gradient descent (GD) even when the starting point $x_0$ is far from the local minimum. For example, consider $f(x) = \min\{1, |x|\}$. Its gradient is zero for any $x \notin [-1, 1]$, which means that GD initialized outside this region will be stationary. Moreover, for large smoothing parameters $\gamma$, RS will tend to flatten the objective and thus lead to the same behavior. However, BMR (with a sufficiently large $\alpha$) will create an almost quadratic function in a region $x \in [-\mu, \mu]$, thus allowing GD to converge even when initialized at distance $\mu$ from the origin.

## 3  Application to first-order and distributed optimization

Using BMR for optimization simply consists in applying first-order methods to the smooth approximation $f_{\gamma,\alpha}$ instead of the objective function $f$. To do so, the optimization algorithm needs to access the (approximate) gradients $\nabla f_{\gamma,\alpha}(x)$ for any $x \in \mathbb{R}^d$. In this section, we first derive an analytical formulation of the gradient of the smoothed function, propose two approximation schemes and discuss the number of samples necessary to reach a sufficient approximation. Then, we use this result to derive novel convergence rates for distributed optimization using BMR smoothing.

### 3.1  Gradient computation and approximation

The gradient of a BMR approximation is surprisingly simple, and a direct extension to that of RS. While $\nabla f_\mu^{RS}(x) = \mathbb{E}\left[\nabla f(x + \gamma X)\right]$ is an expectation of the true gradient in a (Gaussian) neighborhood of $x$, the gradient of a BMR approximation is an expectation of the true gradient over a neighborhood of $x$ *biased towards low values of* $f$. More precisely, let the random variable $Y_{x,\gamma,\alpha} \sim q(y) \propto e^{-\alpha f(y)} p_{x,\gamma}(y)$, where $p_{x,\gamma}$ is the density of a Gaussian distribution $\mathcal{N}(x, \gamma^2 I)$. Intuitively, $Y_{x,\gamma,\alpha}$ is a noisy version of the input $x$ biased towards low values of the function. Then, the following Lemma holds.

**Lemma 1.** *For any function* $f : \mathbb{R}^d \to \mathbb{R}$*, the gradient of its BMR smoothing* $\nabla f_{\gamma,\alpha}$ *is*

$$\nabla f_{\gamma,\alpha}(x) = \mathbb{E}\left[\nabla f(Y_{x,\gamma,\alpha})\right] . \tag{7}$$

In practice, estimating this gradient requires to approximate the expectation in Eq. (7). This may be achieved in multiple ways. We now discuss two approaches: 1) **Langevin Monte-Carlo:** A natural approach to estimate $\nabla f_{\gamma,\alpha}$ consists in sampling the r.v. $Y_{x,\gamma,\alpha}$. While the definition of its density distribution (proportional to $\exp(g(x))$ for a given function $g$) makes sampling relatively hard, Langevin Monte Carlo [25] and MCMC [26] methods are particularly well suited to the sampling of such random variables. However, this approach is difficult to parallelize due to the sequential nature of Langevin Monte Carlo and MCMC methods, and we thus leave the implementation and analysis of such approximation for future work. 2) **Importance sampling:** A second approach consists in sampling another distribution (close to that of $Y_{x,\gamma,\alpha}$) and re-weighting the samples accordingly. For simplicity, we now consider the Gaussian distribution, although more elaborate distributions could lead to substantial improvements in practice. Approximating the gradient can thus be achieved by sampling $K$ noisy inputs $Y_k = x_t + \gamma X_k$, where $X_k \sim \mathcal{N}(0, I)$ for $k \in [\![1, K]\!]$. Then, a weighted average of the gradients $\nabla f(Y_k)$ provide an approximation of the smooth gradient: $\nabla f_{\gamma,\alpha}(x) \approx \sum_k w_k \nabla f(Y_k)$, where $w_k = e^{-\alpha f(Y_k)} / \sum_l e^{-\alpha f(Y_l)}$ for $k \in [\![1, K]\!]$. We discuss the approximation error of this estimation in Section 3.4.

**Remark 2.** Quite surprisingly, BMR smoothing also enjoys a gradient-free approximation of its gradient, allowing to use BMR smoothing for zeroth-order optimization:

$$\nabla f_{\gamma,\alpha}(x) = -\frac{1}{\alpha\gamma^2} \mathbb{E}\left[Y_{x,\gamma,\alpha} - x\right] . \tag{8}$$

For example, applying gradient descent to $f_{\gamma,\alpha}$ with the optimal step-size $\eta = \alpha\gamma^2$ (see Proposition 3) and Eq. (8) gives $x_{t+1} = \mathbb{E}\left[Y_{x_t,\gamma,\alpha}\right]$. The intuition is thus that, at each iteration, points in a Gaussian neighborhood of $x_t$ are sampled and re-weighted to give more importance to low-values of the objective. This scheme is central to evolutionary strategies such as simulated annealing [27], NES [28] or CMA-ES [29], and provide an interesting alternative interpretation of such zeroth-order exploration schemes as first-order optimization of well-chosen smooth approximations. A more detailed analysis of the relationship between exploration and smoothing is left for future work.

---
**Algorithm 1** BMR-GD
---
**Input:** iterations $T$, samples $K$, gradient step $\eta$, smoothing parameter $\gamma$, distortion parameter $\alpha$.
**Output:** optimizer $x_T$
  1: $x_0 = 0$
  2: **for** $t = 0$ to $T - 1$ **do**
  3:      $Y_k = x_t + \gamma X_k$, where $X_k \sim \mathcal{N}(0, I)$ for $k \in [\![1, K]\!]$
  4:      $f^* = \min_k f(Y_k)$
  5:      $w_k = e^{-\alpha(f(Y_k) - f^*)} / \sum_l e^{-\alpha(f(Y_l) - f^*)}$ for $k \in [\![1, K]\!]$
  6:      $G_t = \sum_k w_k \nabla f(Y_k)$
  7:      $x_{t+1} = x_t - \eta G_t$
  8: **end for**
  9: **return** $x_T$
---

**Remark 3.** The gradient of the Moreau envelope (or, more precisely, the proximal operator [30]) is an important tool of non-smooth and composite optimization. Following the definition $\text{prox}_{\mu f} = x - \mu \nabla f_\mu^{ME}(x)$, we may extend the prox operator to $\text{prox}_{f,\gamma,\alpha}(x) = \mathbb{E}\left[Y_{x,\gamma,\alpha}\right]$ . Using Eq. (7) and Eq. (8), we have that $\mathbb{E}\left[Y_{x,\gamma,\alpha}\right] = \mathbb{E}\left[x - \alpha\gamma^2 \nabla f(Y_{x,\gamma,\alpha})\right]$ which can be interpreted as a stochastic equivalent of the formula $\text{prox}_{\mu f}(x) = (I + \mu \nabla f)^{-1}(x)$ that identifies proximal gradient descent to an implicit optimization scheme and is central to its theoretical analysis [30, 19].

## 3.2 Gradient descent scheme

Alg. 1 describes BMR-GD, a (stochastic) gradient descent algorithm on the BMR smooth approximation of the objective. As the gradient $\nabla f_{\gamma,\alpha}(x_t)$ cannot be computed exactly, we approximate it using importance sampling (see Section 3.1). Note that the purpose of step 4 in Alg. 1 is to provide more stability to the computation of the weights $w_k$. The practical performance of this algorithm on non-smooth and non-convex problems is discussed in Section 4.

## 3.3 Distributed optimization

For didactic purposes, let us now show how BMR can improve distributed optimization. In this setting, parallelization can be used to improve the estimation of the gradient by increasing the number of gradient samples $K$. When the number of workers is large, we may, as a first approximation, neglect the gradient estimation error (this error is however discussed in the next section). Then, accelerated gradient descent on a smooth approximation of the objective leads to an approximation error $\mathbb{E}\left[f(x_t)\right] - \min_x f(x) \leq 2A + 2B\|x_0 - x^*\|^2(t + 1)^{-2}$ (see [31, Theorem 3.19]), where $A$ is the approximation error, $B$ is the smoothness, and $x^*$ is an optimum value of the smooth approximation. For RS, we have $A = \mu L\sqrt{d}$ and $B = L/\mu$, and thus, with a proper choice of $\mu$,

$$\mathbb{E}\left[f(x_t)\right] - \min_x f(x) \leq \frac{4L\|x_0 - x^*\|d^{1/4}}{t + 1} . \tag{9}$$

Note that, even with an infinite number of workers (and thus no estimation error on the gradient), the convergence rate depends on the dimension $d$. Using BMR allows to reduce this dependency in the dimension, as we have, with a proper choice of $\mu$,

$$\mathbb{E}\left[f(x_t)\right] - \min_x f(x) \leq \frac{4L\|x_0 - x^*\|}{t + 1} \left(\frac{1}{2} + \sqrt{\frac{d}{\max\{1, \mu L\alpha\}}}\right)^{1/2} . \tag{10}$$

When $\mu L\alpha \approx d$, the dimension-dependent term disappears from the convergence rate, thus improving the optimization of high-dimensional objectives. As discussed in the next section, this improvement in convergence rate is at the cost of a more difficult estimation of the gradient, thus requiring more workers / samples to estimate the gradient within a sufficient precision.

## 3.4 Gradient bias

The main difficulty with Alg. 1 is that the gradient estimator is *biased*. In this section, we quantify this bias and show its dependency in the number of samples. This illustrates the trade-off between

dimensionality of the problem, number of data available to compute the gradient and parameters of the smoothing.

**Proposition 7** (Gradient bias). *Let* $f : \mathbb{R}^d \to \mathbb{R}$ *be a L-Lipschitz function. Then for any* $x \in \mathbb{R}^d$, *any sample size* $K > 0$ *and any* $\varepsilon > 0$, *the estimate bias is bounded as follows:*

$$\left\| \mathbb{E} \left[ \hat{\nabla}_K f_{\gamma,\alpha}(x) \right] - \nabla f_{\gamma,\alpha}(x) \right\| \leq \frac{c_\varepsilon L \sigma_x}{K^{\frac{1}{2}-\varepsilon}} \left( 1 + \frac{\sigma_x}{K^{\frac{1}{2}-\varepsilon}} \right), \tag{11}$$

*where* $c_\varepsilon > 0$ *and* $\sigma_x = \mathrm{std}\left( e^{-\alpha f(x+\gamma X)} \right) / \mathbb{E}\left[ e^{-\alpha f(x+\gamma X)} \right]$ *is the normalized standard deviation of* $e^{-\alpha f(x+\gamma X)}$.

This result shows that the estimate bias is almost in $O\left(1/\sqrt{K}\right)$. The quality of the estimate is impacted by the variations of the function around the current point, in particular if the largest variations are concentrated in a small area, since a high value of $\alpha$ will make the estimate focus only on a few sample points. Although the quality of the estimator could be degraded if a point is drawn where the function is significantly lower, this could lead to a better descent direction in practice. Finally, let us point out that the constant term $c_\varepsilon$ is very sensitive to changes in the method parameters, in particular $\alpha$. The detailed expression of $c_\varepsilon$ may be found in the supplementary material.

## 4 Empirical Results

The goal of this section is to go beyond theoretical results and investigate how BMR properties impact the derived optimization scheme. Computing the ME being too expensive in practice, we decided to benchmark BMR mainly with RS.

It is worth mentioning that BMR can be easily distributed over a large number of workers in a similar fashion than RS, making them competitive w.r.t. classical GD which can't be distributed in this setting (see Section 3.3). However, we decided to add GD as an example of local minima achieved with such methods. Furthermore we select the learning rate, used for BMR and RS, based on GD performances.

We benchmarked our approach on synthetic non-smooth functions that we believe are representative of struggling problems arising with complex real world loss functions and deep learning losses. Experiments being proofs of concepts, parameters are chosen big enough to distinguish among the different methods but small enough for ease of computation, letting extensions to large scale deep learning problems for future work. Optimization is thus done in a 10-dimensional space and 10 data points are sampled at each iteration of the algorithms. We test the methods with $(\gamma, \alpha) \in \{0.01, 0.1, 1, 10, 100\}^2$, we display only few of them for seek of readability but conclusion remains the same for other values. In the following, optimization schemes are benchmarked using the ratio $\frac{f(x_0) - f(x_t)}{f(x_0) - f(x^\star)}$ as the performance criterion.

### 4.1 Approximation error and $\gamma$ sensitivity

First, we confirm that BMR smoothing achieves indeed in practice a smaller approximation error than RS, thanks to its $\alpha$ parameter which allows to go from a $\mu L \sqrt{d}$ error to a $\mu L$ one as $\alpha$ grows. This non desirable behaviour of RS is outlined in Figure 1a, where the minimum of the smoothed function does not align with the minimum of the original function. In this first experiment, we emphasize this difference by showing that even in the simple case of a convex function and for the same amount of noise used in the smoothing operator, BMR can achieve the minimum of the function while RS cannot. To this end, we consider the function $f_R(x, y) = \|x\|_2 + \sum_i g(y_i)$ where $g(y_i) = y_i$ if $y_i > 0$ and $-y_i/10$ otherwise. This simple, yet talkative, example of non-smooth and non-symmetric function arise in deep learning losses using activation units such as ReLU variants.

Figure 2 shows that RS is falling in minimizing the task when $\gamma = 10$, while BMR, even with a relatively small $\alpha$, reaches the optimum, what makes the optimization scheme less sensitive to the choice of $\gamma$. Even if decreasing $\gamma$ solves the issue for RS, it implies a more refined hyperparameter search and brings out the impact of BMR approximation error reduction.

We thus investigate the previously highlighted $\gamma$ amplitude sensitivity. Following Proposition 2, we coupled the evolution of $\alpha$ and $\gamma$ by using a $\gamma$ exponential decay schedule and set $\alpha \propto \gamma^{-2}$ in order

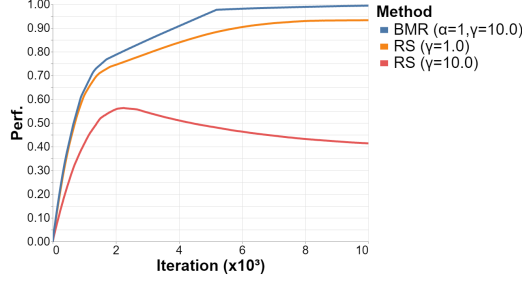

Figure 2: Performances of BMR and RS while minimizing $f_R$ with different $\gamma$s

to smooth border effects. For this experiment we optimize a piecewise linear function constructed by adding a fixed positive noise $m$ to $x \mapsto \|x\|_1$. For each component, we select a set of points $a_{-n} < .. < a_0 = 0 < .. < a_n$ and construct their respective images $(b_i)_{i \in [-n..n]}$ where $b_{2i} = \|a_{2i}\|$ and $b_{2i+1} = \|a_{2i}\| + m_i$. We then construct a piecewise linear function and denote it $f_{PWL}$.

Results are reported in Figure 3 which outlines the following advantages of BMR over RS: BMR is less dependent than RS on the $\gamma$ parameter tuning and it finds better optima than RS. We notice however that choosing a $\gamma$ too small makes either RS and BMR to converge to the same local minima as GD, due to the lack of exploration.

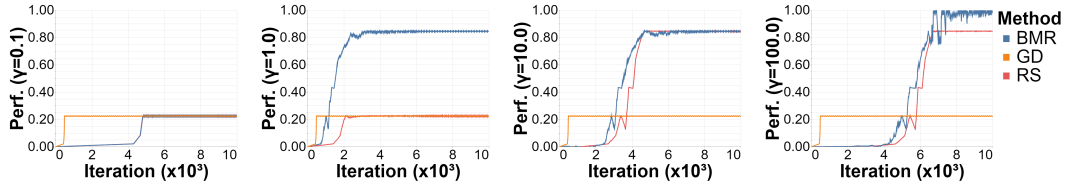

Figure 3: $\gamma$-decaying schedule with different initial values while minimizing $f_{PWL}$.

First synthetic experiments tend to show that the use of a scheduler ease the choice of $\gamma$ paying a low time cost with no performance loss.

## 4.2 An $\alpha$-opportunist method

In Section 2.3, we have seen that theoretically, other things being equal, increasing $\alpha$ should widen local minima, the deepest ones being the widest. We see in this section that in practice, BMR indeed converges quicker to local minima and better ones as $\alpha$ increases. To this end, we consider *Fourier* inspired functions: let $f_F(x) = \sum_{i,j} a_j cos(b_j x_i)$, where $(c_j, b_j)_{j \in [1..10]}$ are chosen randomly, and $f_{FC}(x) = \min(1, f_F(x))$.

Results are reported in Figure 4. First, we see that for both $f_F$ and $f_{FC}$, larger $\alpha$ leads to better performance. At convergence, gradients around the reached local minima are also smaller and more stable. Second, larger $\alpha$ are able to go against local gradients to reach the best local minima in the neighborhood: in both examples, $\alpha = 100$ clearly degrades its performances to reach better local minima than $\alpha = 10$. The larger the $\alpha$ the more opportunist the method while RS tends to smoothly increase the performances.

Converging to the nearest local minimum can however bring unwanted behavior. We can see for instance on $f_{FC}$ that BMR gets trapped in a decent local optimum but RS leads to better performance in the long run. In such case, looking farther by increasing $\gamma$ may be necessary. Finding good heuristics to smoothly transition from RS to ME (i.e. "$+\infty$"-BMR), and trying to get the best of both methods is left for future work.

## 5 Conclusion

In this paper, we have introduced a novel smoothing procedure called BMR interpolating between RS and ME. Theoretically, we have shown that this method was able to escape the $\sqrt{d}$-dependency

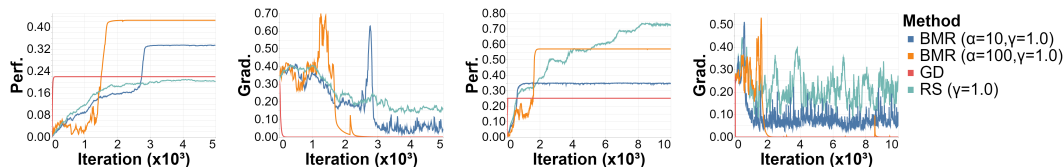

Figure 4: Performance and corresponding gradient while minimizing $f_{\mathrm{F}}$ (left) and $f_{\mathrm{FC}}$ (right).

of RS approximation error, while working on non-convex functions as opposed to ME. From a computational perspective, we have seen that BMR is easily computable contrary to ME and even if it comes with an additional hyperparameter $\alpha$ compared to RS, it eases the selection of the noise amplitude $\gamma$ by being more robust to it.

A promising application of this method would concern the Reinforcement Learning field, where a set of methods, called Policy Gradient methods, apply first-order optimization techniques on a non-convex loss function. Those methods are known to be prone to converge towards local optima and would be thus a candidate of choice for applying BMR.

## Broader Impact

As a method accelerating the optimization of non-smooth and non-convex functions, the BMR smoothing operator may be applied to a wide range of applications, in particular for training deep neural networks of which the induced loss function are highly non-smooth and non-convex and of which applications are numerous: image, speech, natural language processing, reinforcement learning... As a theoretical work however, it remains difficult to more precisely evaluate the impact of this paper. Exhaustive experimental benchmarks assessing the performance of BMR smoothing should first be conducted.

## Acknowledgments and Disclosure of Funding

The authors thank the whole team at Huawei Paris and in particular Aladin Virmaux and Cédric Malherbe for useful discussions. This work was financed by Huawei Technologies.

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
