[Supplementary Material]

# Supplementary Material

This document contains the proofs of the results presented in the paper: A Simple and Efficient Smoothing Method for Faster Optimization and Local Exploration.

**Proof of Proposition 1.** These two properties directly follow from the formulation $f_{\gamma,\alpha}(x) = -\frac{1}{\alpha} \ln \left( \mathbb{E} \left[ e^{-\alpha f(x+\gamma X)} \right] \right)$.

**Proof of Proposition 2.** First, note that $\phi_0(x) = x$, and thus we have $f_{\mu,\alpha}(x) = \phi_\alpha^{-1} \left( \mathbb{E} \left[ \phi_\alpha \left( f(x+\mu X) \right) \right] \right) \to_{\alpha \to 0} \mathbb{E} \left[ f(x+\mu X) \right] = f_\mu^{RS}(x)$. Then, we derive the behavior when $\alpha$ tends to $+\infty$ as follows:

$$
\begin{aligned}
f_{\sqrt{\frac{\mu}{\alpha}},\alpha}(x) &= & -\frac{1}{\alpha} \ln \left( \mathbb{E} \left[ e^{-\alpha f(x+\frac{\mu}{\alpha}X)} \right] \right) \\
&= & -\frac{1}{\alpha} \ln \left( \int_{y \in \mathbb{R}^d} e^{-\alpha f(x+y) - \frac{\alpha \|y\|^2}{2\mu}} C_{\mu,\alpha} dy \right) \\
&= & -\frac{1}{\alpha} \ln C_{\mu,\alpha} - \ln \left\| e^{-f(x+y) - \frac{\|y\|^2}{2\mu}} \right\|_\alpha \\
&\to_{\alpha \to +\infty} & -\ln \left( \max_{y \in \mathbb{R}^d} e^{-f(x+y) - \frac{\|y\|^2}{2\mu}} \right) \\
&= & \min_{y \in \mathbb{R}^d} f(x+y) + \frac{\|y\|^2}{2\mu} \\
&= & f_\mu^{ME}(x),
\end{aligned}
\tag{12}
$$

where $C_{\mu,\alpha} \propto \left( \frac{\alpha}{\mu} \right)^{d/2}$ is the density of $\mathcal{N} \left( 0, \sqrt{\frac{\mu}{\alpha}} I \right)$ at $x=0$, and the $L_p$-norm is taken with respect to the Lebesgue measure in $\mathbb{R}^d$.

**Proof of Proposition 3.** As discussed in Section 2, $f_{\gamma,\alpha}(x) = -\frac{1}{\alpha} \ln \left( e^{-\alpha f} * p_\gamma \right)(x)$, and using the derivation formula for the convolution: $(f * g)' = f' * g = f * g'$, we have

$$
\partial_i \partial_j f_{\gamma,\alpha}(x) = \frac{1}{\alpha \gamma^2} \left[ \frac{\mathbb{E} \left[ e^{-\alpha f(x+\gamma X)} (\mathbb{1}\{i=j\} - X_i X_j) \right]}{\mathbb{E} \left[ e^{-\alpha f(x+\gamma X)} \right]} + \frac{\mathbb{E} \left[ e^{-\alpha f(x+\gamma X)} X_i \right] \mathbb{E} \left[ e^{-\alpha f(x+\gamma X)} X_j \right]}{\mathbb{E} \left[ e^{-\alpha f(x+\gamma X)} \right]^2} \right].
\tag{13}
$$

Let $u \in \mathbb{R}^d$. Summing over $i,j$ the above terms leads to

$$
\begin{aligned}
u^\top \nabla^2 f_{\gamma,\alpha}(x) u &= & \sum_{ij} u_i u_j \partial_i \partial_j f_{\gamma,\alpha}(x) \\
&\leq & \frac{1}{\alpha \gamma^2} \left[ \|u\|^2 - \frac{\mathbb{E}\left[ e^{-\alpha f(x+\gamma X)} \langle X,u \rangle^2 \right]}{\mathbb{E}\left[ e^{-\alpha f(x+\gamma X)} \right]} + \frac{\mathbb{E}\left[ e^{-\alpha f(x+\gamma X)} \langle X,u \rangle \right]^2}{\mathbb{E}\left[ e^{-\alpha f(x+\gamma X)} \right]^2} \right] \\
&\leq & \frac{1}{\alpha \gamma^2} \left[ \|u\|^2 - \frac{1}{\gamma^2} \operatorname{var}(\langle Y_{x,\gamma,\alpha}, u \rangle) \right] \\
&\leq & \frac{1}{\alpha \gamma^2} \|u\|^2,
\end{aligned}
\tag{14}
$$

where $Y_{x,\gamma,\alpha} \sim q(y) \propto e^{-\alpha f(y)} p_{x,\gamma}(y)$ and $p_{x,\gamma}$ is the density of a Gaussian distribution $\mathcal{N}(x,\gamma^2 I)$.

**Proof of Proposition 4.** First, note that, for any random variable $X \in \mathbb{R}$, we have $\ln \mathbb{E} \left[ e^X \right] \geq -\ln \mathbb{E} \left[ e^{-X} \right]$. This result follows from Jensen's inequality on the convex function $x \mapsto 1/x$ (for $x > 0$), as then $\mathbb{E} \left[ e^{-X} \right] = \mathbb{E} \left[ 1/e^X \right] \geq 1/\mathbb{E} \left[ e^X \right]$ and taking the logarithm leads to the desired inequality. As a direct consequence, we also have that

$$
\left| \ln \mathbb{E} \left[ e^X \right] \right| \leq \ln \mathbb{E} \left[ e^{|X|} \right],
\tag{15}
$$

using $-|X| \leq X \leq |X|$ and the monotonicity of the logarithm and exponential functions. Finally, by Proposition 1, BMR smoothing is translation equivariant, and taking $f(x)$ as a constant,

$$
\begin{aligned}
|f_{\gamma,\alpha}(x) - f(x)| &= |(f - f(x))_{\gamma,\alpha}(x)| \\
&= \frac{1}{\alpha} \left| \ln \mathbb{E} \left[ e^{-\alpha(f(x+\gamma X) - f(x))} \right] \right| \\
&\leq \frac{1}{\alpha} \ln \mathbb{E} \left[ e^{\alpha |f(x+\gamma X) - f(x)|} \right] \\
&\leq \frac{1}{\alpha} \ln \mathbb{E} \left[ e^{\alpha \gamma L \|X\|_2} \right],
\end{aligned}
\tag{16}
$$

where the first inequality uses the alternative definition of BMR smoothing $f_{\gamma,\alpha}(x) = -\frac{1}{\alpha} \ln \left( \mathbb{E} \left[ e^{-\alpha f(x+\gamma X)} \right] \right)$, and the second inequality uses the Lipschitz assumption on $f$. The last term is the logarithm of the moment generating function of a Chi distribution of parameter $d$, which is 1-sub-gaussian [32]. Hence, we have, $\forall t \geq 0$,

$$
\mathbb{E} \left[ e^{t\|X\|_2} \right] \leq e^{\frac{t^2}{2} + t\mathbb{E}[\|X\|_2]} \leq e^{\frac{t^2}{2} + t\sqrt{d}},
\tag{17}
$$

leading to the desired result with $t = \alpha\gamma L$.

**Proof of Proposition 5.** First, note that, as $f(x) \geq f(x^*)$ for any $x \in \mathbb{R}^d$, we also have $f_{\gamma,\alpha}(x) \geq f(x^*)$. Then, as $f$ is $L$-Lipschitz, we have, $\forall c > 0$, $f(x+\gamma X) - f(x) \leq \gamma L\|X\| \leq \gamma L(c\|X\|^2 + \frac{1}{4c})$, where the second inequality follows from the convexity of $x \mapsto x^2$. Moreover, $\|X\|^2$ follows a $\chi^2(d)$ distribution, and $\mathbb{E} \left[ e^{t\|X\|^2} \right] = (1 - 2t)^{d/2}$. Finally, we have

$$
\begin{aligned}
f_{\gamma,\alpha}(x) - f(x) &= -\frac{1}{\alpha} \ln \left( \mathbb{E} \left[ e^{-\alpha(f(x+\gamma X) - f(x))} \right] \right) \\
&\leq -\frac{1}{\alpha} \ln \left( \mathbb{E} \left[ e^{-\alpha\gamma L(c\|X\|^2 + \frac{1}{4c})} \right] \right) \\
&\leq \frac{\gamma L}{4c} - \frac{1}{\alpha} \ln \left( \mathbb{E} \left[ e^{-\alpha\gamma Lc\|X\|^2} \right] \right) \\
&= \frac{\gamma L}{4c} + \frac{d}{2\alpha} \ln \left( 1 + 2\alpha\gamma Lc \right).
\end{aligned}
\tag{18}
$$

Chosing $c = \frac{\alpha\gamma L}{2d}$ concludes the proof.

**Proof of Proposition 7.** Let $x \in \mathbb{R}^d$. One aims at estimating the gradient of $f_{\gamma,\alpha}$ at $x$, that is

$$
\nabla f_{\gamma,\alpha}(x) = \mathbb{E} \left[ \frac{e^{-\alpha f(Y)}}{\mathbb{E} \left[ e^{-\alpha f(Y')} \right]} \nabla f(Y) \right],
$$

where $Y$ and $Y'$ are independent random vectors drawn from $\mathcal{N}(0, \gamma I_d)$. Let $Y_1, \ldots, Y_K$ be i.i.d. random vectors drawn from $\mathcal{N}(0, \gamma I_d)$. We define the following estimate:

$$
\hat{\nabla}_K f(x) \triangleq \frac{1}{K} \sum_{k=1}^{K} \frac{e^{-\alpha(f(Y_k) - f(x))}}{T_K} \nabla f(Y_k),
$$

where $T_K \triangleq (1/K) \sum_{k=1}^{K} e^{-\alpha(f(Y_k) - f(x))}$ is the renormalization factor. Using the fact that $Y, Y', Y_1, \ldots, Y_K$ are i.i.d. one can write the estimate bias as follows:

$$
\begin{aligned}
\left\| \mathbb{E} \left[ \hat{\nabla}_K f_{\gamma,\alpha}(x) \right] - \nabla f_{\gamma,\alpha}(x) \right\| &= \left\| \mathbb{E} \left[ \frac{1}{K} \sum_{k=1}^{K} \frac{e^{-\alpha(f(Y_k) - f(x))}}{T_K} \nabla f(Y_k) \right] - \mathbb{E} \left[ \frac{e^{-\alpha(f(Y) - f(x))}}{\mathbb{E} \left[ e^{-\alpha(f(Y') - f(x))} \right]} \nabla f(Y) \right] \right\| \\
&= \left\| \mathbb{E} \left[ \frac{e^{-\alpha(f(Y_1) - f(x))}}{T_K} \nabla f(Y_1) \right] - \mathbb{E} \left[ \frac{e^{-\alpha(f(Y) - f(x))}}{\mathbb{E} \left[ T_K \right]} \nabla f(Y) \right] \right\| \\
&= \left\| \mathbb{E} \left[ e^{-\alpha(f(Y_1) - f(x))} \nabla f(Y_1) \left( \frac{1}{T_K} - \frac{1}{\mathbb{E} \left[ T_K \right]} \right) \right] \right\|.
\end{aligned}
$$

The quality of the gradient estimate thus revolves around the gap between the two ratios in the RHS. This gap may be tricky to upper bound if some values of the function $f$ are very small in comparison to $f(x)$, so we will try and characterize the gap using the variance of $f$ around $x$.

For any $t > 0$, we consider the event $\mathcal{E}_{x,K}(t) \triangleq \{|f(Y_k) - f(x)| \leq t,\ 1 \leq k \leq K\}$. Using the convexity and the Taylor expansion of $x \mapsto 1/x$ around $\mathbb{E}\left[T_K | \mathcal{E}_{x,K}(t)\right]$, one can check that under $\mathcal{E}_{x,K}(t)$ the following holds

$$\frac{1}{\mathbb{E}\left[T_K\right]} - \frac{T_K - \mathbb{E}\left[T_K\right]}{\left(\mathbb{E}\left[T_K\right]\right)^2} \leq \frac{1}{T_K} \leq \frac{1}{\mathbb{E}\left[T_K\right]} - \frac{T_K - \mathbb{E}\left[T_K\right]}{\left(\mathbb{E}\left[T_K\right]\right)^2} + e^{\alpha t}\frac{\left(T_K - \mathbb{E}\left[T_K\right]\right)^2}{\left(\mathbb{E}\left[T_K\right]\right)^2},$$

and taking the conditional expectation w.r.t. $\mathcal{E}_{x,K}(t)$ yields:

$$\mathbb{E}\left[\left|\frac{1}{T_K} - \frac{1}{\mathbb{E}\left[T_K\right]}\right|\,\middle|\,\mathcal{E}_{x,K}(t)\right] \leq \frac{\mathbb{E}\left[|T_K - \mathbb{E}\left[T_K\right]|\,\big|\,\mathcal{E}_{x,K}(t)\right]}{\left(\mathbb{E}\left[T_K\right]\right)^2} + e^{\alpha t}\frac{\mathbb{E}\left[(T_K - \mathbb{E}\left[T_K\right])^2|\mathcal{E}_{x,K}(t)\right]}{\left(\mathbb{E}\left[T_K\right]\right)^2}$$

$$\leq \frac{\mathbb{E}\left[|T_K - \mathbb{E}\left[T_K\right]|\right]}{\mathbb{P}\left(\mathcal{E}_{x,K}(t)\right)\left(\mathbb{E}\left[T_K\right]\right)^2} + e^{\alpha t}\frac{\mathbb{E}\left[(T_K - \mathbb{E}\left[T_K\right])^2\right]}{\mathbb{P}\left(\mathcal{E}_{x,K}(t)\right)\left(\mathbb{E}\left[T_K\right]\right)^2}$$

$$\leq \frac{\sqrt{\mathrm{var}\left(T_K\right)}}{\mathbb{P}\left(\mathcal{E}_{x,K}(t)\right)\left(\mathbb{E}\left[T_K\right]\right)^2} + e^{\alpha t}\frac{\mathrm{var}(T_K)}{\mathbb{P}\left(\mathcal{E}_{x,K}(t)\right)\left(\mathbb{E}\left[T_K\right]\right)^2}$$

$$= \frac{\sqrt{\mathrm{var}\left(T_1\right)}}{\sqrt{K}\mathbb{P}\left(\mathcal{E}_{x,K}(t)\right)\left(\mathbb{E}\left[T_1\right]\right)^2} + e^{\alpha t}\frac{\mathrm{var}(T_1)}{K\mathbb{P}\left(\mathcal{E}_{x,K}(t)\right)\left(\mathbb{E}\left[T_1\right]\right)^2}.$$

We can now upper bound the bias of our estimator:

$$\left\|\mathbb{E}\left[\hat{\nabla}_K f_{\gamma,\alpha}(x)\right] - \nabla f_{\gamma,\alpha}(x)\right\| \leq Le^{\alpha t}\left(\frac{\sqrt{\mathrm{var}\left(T_1\right)}}{\sqrt{K}\left(\mathbb{E}\left[T_1\right]\right)^2} + e^{\alpha t}\frac{\mathrm{var}(T_1)}{K\left(\mathbb{E}\left[T_1\right]\right)^2}\right) + 2L(1 - \mathbb{P}\left(\mathcal{E}_{x,K}(t)\right)).$$

Since $f$ is $L$-Lipschitz, standard concentration results on $\mathcal{E}_{x,K}(t)$ yields:

$$\mathbb{P}\left(\mathcal{E}_{x,K}(t)\right) = \mathbb{P}\left(\mathcal{E}_{x,1}(t)\right)^K \geq \left(1 - 2e^{-\frac{t^2}{2L^2\gamma^2}}\right)^K$$

We can now bound the bias as follows

$$\left\|\mathbb{E}\left[\hat{\nabla}_K f_{\gamma,\alpha}(x)\right] - \nabla f_{\gamma,\alpha}(x)\right\| \leq \inf_{t>0} Le^{\alpha t}\left(\frac{\sigma_x}{\sqrt{K}e^{-\alpha(f_{\gamma,\alpha}(x)-f(x))}} + e^{\alpha t}\frac{\sigma_x^2}{K}\right) + 2L\left(1 - \left(1 - 2e^{-\frac{t^2}{2L^2\gamma^2}}\right)^K\right)$$

$$\leq \inf_{t>0} Le^{\alpha t}\left(\frac{\sigma_x}{\sqrt{K}e^{-\alpha(f_{\gamma,\alpha}(x)-f(x))}} + e^{\alpha t}\frac{\sigma_x^2}{K}\right) + 4KLe^{-\frac{t^2}{2L^2\gamma^2}}.$$

Let $c > 0$, picking $t = L\gamma\sqrt{3\log(cK)}$ yields:

$$\left\|\mathbb{E}\left[\hat{\nabla}_K f_{\gamma,\alpha}(x)\right] - \nabla f_{\gamma,\alpha}(x)\right\| \leq \frac{L}{\sqrt{K}}\left(\sigma_x e^{\alpha L\gamma\sqrt{3\log(cK)}+\alpha(f_{\gamma,\alpha}(x)-f(x))} + 4c^{-3/2}\right) + \frac{L\sigma_x^2}{K}e^{2\alpha L\gamma\sqrt{3\log(cK)}}.$$

Let $x_0 > 0$, by concavity of $x \mapsto \sqrt{x}$, we can write:

$$\sigma_x e^{\alpha L\gamma\sqrt{3\log(cK)}} \leq \sigma_x e^{\frac{\alpha L\gamma}{2}\sqrt{x_0}}e^{\frac{3\alpha L\gamma}{2\sqrt{x_0}}\log(cK)}$$

$$= \sigma_x e^{\frac{\alpha L\gamma}{2}\sqrt{x_0}}(cK)^{\frac{3\alpha L\gamma}{2\sqrt{x_0}}}.$$

Let $\varepsilon \triangleq 3\alpha L\gamma/(2\sqrt{x_0})$. The upperbound can be rewritten as follows:

$$\left\|\mathbb{E}\left[\hat{\nabla}_K f_{\gamma,\alpha}(x)\right] - \nabla f_{\gamma,\alpha}(x)\right\| \leq \frac{L}{\sqrt{K}}\left(\sigma_x e^{\frac{\alpha L\gamma}{2}\sqrt{x_0}+\alpha(f_{\gamma,\alpha}(x)-f(x))}(cK)^\varepsilon + 4c^{-3/2}\right) + \frac{L\sigma_x^2}{K}(cK)^{2\varepsilon},$$

and the result holds.