[Reviews · NeurIPS 2020]

Review 1

Summary and Contributions: This work proposes the “Bend, Mix, and Release” technique, a smoothing method that extends the previously studied smoothing techniques of randomized smoothing and Moreau-envelope smoothing, whereby it provides a tradeoff between the better approximation error of Moreau-envelope, with the better computational cost of randomized smoothing, based on a certain parameterizing term alpha.

Strengths: The paper introduces a new notion of smoothing which neatly interpolates between the regimes of randomized smoothing and Moreau-envelope smoothing, which as far as I can tell, appears to be a novel observation. The smoothed function, based on a certain “bending function” phi, has several interpretations including from the perspective of moment generating functions, as a Gaussian filter, and in terms of the negative log of the lp norm of the exponentiation of a randomized variant of the function. The function is parameterized by some alpha, where as alpha -> infinity, the function approaches the Moreau-envelope smoothing. The catch is that calculating the gradient of this function (which is needed for first-order optimization on the smooth surrogate) requires a sampling procedure, which the paper also details further, and which can become more expensive as alpha increases. Overall, this work provides both a theoretically appealing interpolation between randomized smoothing and Moreau-envelope smoothing, as well as potential for practitioners who may hope to work with more nuanced ways of inducing smoothness, and so I believe it would be a nice addition to the conference.

Weaknesses: ===== Minor comments ===== - While the title includes "Accelerated Optimization", it is not clear that this technique is specifically for acceleration in the standard sense in the optimization literature (i.e., Nesterov's acceleration), so this choice of wording either requires additional explanation, or should otherwise be changed. - A more in-depth discussion/literature review is in order regarding the comments on "shallow" vs "deep" local minima, as this remains a highly debated topic.

Correctness: The method appears correct.

Clarity: The paper is well written.

Relation to Prior Work: Mostly, however, minor comments: - First paragraph, it is unfair to say that (plain) stochastic gradient descent has poor generalization capabilities, given that in fact in practice it performs quite well in terms of generalization. This should be worded more carefully. - Important missing references for smoothing: Nesterov, Yu. "Smooth minimization of non-smooth functions." Mathematical programming 103, no. 1 (2005): 127-152. Beck, Amir, and Marc Teboulle. "Smoothing and first order methods: A unified framework." SIAM Journal on Optimization 22, no. 2 (2012): 557-580.

Reproducibility: Yes

Additional Feedback: After rebuttal: I have read the response and appreciate the authors' clarification for my concerns, so I maintain my score.


Review 2

Summary and Contributions: The paper proposes a smoothing approach, called BMR, for the optimization of non-smooth and potentially non-convex functions. The basic idea is to solve a non-smooth unconstrained optimization problem by replacing the objective with a more convenient “smoothed” objective, which potentially leads to improved convergence. The paper shows that BMR has an improved approximation error, widens local minima, and is amenable for distributed optimization. The technique is tested in numerical experiments.

Strengths: - the approach implicitly generalizers existing techniques (RS, ME). Table I gives a great overview of how the proposed technique compares with related smoothing methods. - the presentation of the BMR smoothing in Section 2 is accessible and well justified. - the analysis of the smoothness and approximation properties of BMR in Section 2.3 are sound. - the result on the widening of local minima is very interesting. - the importance sampling approach of Section 3.1 is also reasonable, but I would have loved to see more justifications for eq (7). - Algorithm 1 is nice and simple. Any rules on the choice of step size to ensure at least local convergence? - In general, the technical part is sound, interesting, and comprehensive. - The results show improved convergence of BMR in simple numerical examples. - the discussion of limitations across the paper and at the end of the experimental section is very useful for the reader.

Weaknesses: - The introduction would benefit from stating the optimization problem the approach solves and the assumptions on the objective (currently, line 32 abruptly mentions “f” without even defining it). The introduction should also include an informal definition of the approximation error used in Table 1. - Experiments: benchmarking only against BR seems restrictive. Adding the comparison against ME (in terms of accuracy and timing) would have been helpful. - Experiments: it would have been great to see more realistic experiments to support the claims that BMR can benefit optimization for ML and deep learning. Currently, seeing a 10-dimensional problem with a toy objective is a bit disappointing. In particular, since the approximation error in RS increases with the dimension, it would have been great to at least see tests with increasing dimension “d”. - after eq (10), the paper says that the bias is almost in O(sqrt(K)), which seems incorrect. Am I missing something? - BMR seems to have a lot of oscillations in Fig 3. Is it expected?

Correctness: The claims and method look correct and the empirical evaluation is sound (but can be extended).

Clarity: The paper is well written.

Relation to Prior Work: yes, the paper is positioned as a generalization of two related works.

Reproducibility: Yes

Additional Feedback: POST REBUTTAL: After reading the reviews and the rebuttal, I remain convinced that this is a very good paper. The only issue I see is that the authors preferred not to add further experimental analysis, which was a source of concern for most reviewers. For this reason, my assessment is between "accept" and "strong accept".


Review 3

Summary and Contributions: The authors claim that they proposed a novel smoothing method.

Strengths: A new smoothing method is proposed based on two existing results. This work may help in several nonsmooth training problems.

Weaknesses: The author combines two exisiting works. They did not show much how smoothing impacts the ML. The numerics are too weak. I dis not see any training problems.

Correctness: It may be correct. The proofs are really simple.

Clarity: Not very well.

Relation to Prior Work: They just combine previous works.

Reproducibility: Yes

Additional Feedback: The authors propose a novel smoothing method. This paper is not well written. The authors combine two exiting works to present their method. First, they do not present enough motivations about how the smoothing method is used for ML. In fact, they just say it may work in the regularized training. 2. The function needs to be Lipschitz, it seems that the methods just can applied to functions that ``not very bad". It seems that this assumption cannot be removed. Can this assumption hold in the neural network training? 3. The numerics are too weak for me. Due to the trivial theoretical contributions, at least, it need to be applied several useful cases. Several network training tasks can be considered in the numerics. 4. The authors put too many Theorems in such a short paper. Most of them are actually lemmas and not the main conclusions. The authors need to revise this. 5. Several typos need to be corrected. ``Notations " is ``notation". You have a ``?" in your proofs in the. ------------------------------------------------------------------------------------------------- comments after rebuttal ------------------------------------------------------------------------------------------------- I have read your feedback. I may not understand the contribution of this paper. I raise the score to 5.


Review 4

Summary and Contributions: This paper describes a way to smooth functions that interpolates between the Moreau envelope and the "randomized sampling" smoothing which approximates a function f with fRS(x) = E_z[f(x+gamma z)] where is a standard Gaussian and gamma is a smoothing parameter. Such an approach is useful because many optimization methods only apply to smooth functions, but can be extended to nonsmooth functions with controlled error by using such smoothings. The key claimed drawback with random sampling is that it introduces an approximation error for a given level of smoothing that is dimension-dependent (on the order of sqrt(d)). The key claimed drawback with the Moreau envelope is that it is difficult to compute (as it involves solving an optimization problem). The proposed interpolation essentially replaces the minimization problem in the Moreau envelope with a "soft" approximation. In particular it uses the approximation min_{x} h(x) approx -alpha^{-1}log [integral_y e^{-alpha h(y)}\;dmu(y)] for an appropriately chosen measure (in this case Lesbegue), where h is the objective function in the definition of the Moreau envelope. (The quadratic term in the Moreau envelope naturally leads to the Gaussian perturbation in the definition of the propsed smoothing given in the paper.) Basic properties of the proposed smoothing are studied, including -- approximation error -- approximation error at a minimizer Furthermore, some suggestions about how to approximately compute the gradient (via sampling) are given, and some analysis is performed on the bias of the resulting gradient estimates.

Strengths: The key contributions is to propose a way to smooth functions that interpolates between the Moreau envelope and the randomized smoothing approaches, in such a way that the approximation error for a given smoothness interpolates between the two. The proposed method appears natural, and is probably novel (although I am not sufficiently knowledgeable about the current literature to judge). A good smoothing method can have a very significant impact on how nonsmooth objective functions are minimized in practice (e.g., in training neural networks). As such, if further investigation show that this method of smoothing works well in practice, then this paper could have quite an impact for the NeurIPS community.

Weaknesses: Although the approach shows promise, the empirical evaluation presented is not particularly thorough. While very detailed numerical studies are not always the norm for a paper of this type, I would find it more convincing that this is a good way to interpolate between the Moreau Envelope and Randomized Sampling approaches to smoothing (not just one way to do so) the more carefully presented the numerical experiments. Some of the claims about "widening local minima" are not really substantiated or even made precise. What does it mean, exactly, to widen a local minimum? The only precise result that the discussion related to this point is associated with is the error bound at a (global) minimum given in Proposition 5.

Correctness: The main claims of the paper seem correct (and are certainly plausible). Since essentially no proofs are supplied in the main paper, and the ideas of the proof are not sketched in the main paper, it is difficult to be sure without a very detailed consultation of the supplementary material.

Clarity: The paper is quite clear overall. It would be nice to outline the key ideas of the proof for the reader for the most important results.

Relation to Prior Work: There is a clear delineation between this work and the key smoothing approaches it seeks to interpolate between. The work is also well situated in the broader context in the opening paragraphs.

Reproducibility: Yes

Additional Feedback: Annealed importance sampling may make sense for gradient evaluation, with the annealing parameter being the smoothing parameter (start with a very smooth version, and then sucessively refine the estimates using smaller and smaller smoothing parameters until reaching the target). It may be helpful to mention that the "randomized sampling" approach to smoothing is (after a minor rescaling) occasionally called the Gaussian noise operator (see Section 11.1 of O'Donnell "Analysis of Boolean Functions") or, again after a further reparametrization, much more standardly called the Ornstein–Uhlenbeck semigroup. It would be helpful if this terminology were adopted to help the community leverage the many things known about this operator. This approach can be generalized by taking any approximation to the Moreau envelope that is more computationally attractive. Indeed essentially what is going on here is that the min in the Moreau envelope is replaced with a "soft-min". What other natural ways are there to approximate the Moreau envelope that could be used in this context?

[Author Response · NeurIPS 2020]

We would like to thank all four reviewers for the close look given at the paper. We believe that the paper gained clarity and readability by taking their comments into account. We now answer the different issues and comments.

**Introduction (REVIEWER 1 AND 2)** *Accelerated optimization*: this choice of wording was made to emphasize the fact that optimizing a smoothed version of the non-smooth objective function would lead faster to a (better) result. However, following your advice and to avoid confusion with the sense commonly given in the field, we will replace this term by *faster optimization*.

*Problem formulation*: we will introduce the problem tackled more formally in the introduction, introducing the objective function $f$ and describing Table 1 as the error bounds on the norm of the difference between the objective function and its resulted smoothed form.

*Poor generalization capabilities*: our sentence is misleading, "resulting in poor generalization capabilities [8]" will be deleted.

**Widening of local minima (REVIEWER 4)** The widening of local minima was indeed not sufficiently developed, and we have thus decided to add the following result to Sec. 2.3 (using the extra page of the camera-ready version).

**Proposition 1** (Widening of local minima)**.** *Let $\mu > 0$ and $y \in \mathbb{R}^d$. If $f$ is L-Lipschitz, then there exists $z \in \mathbb{R}^d$ such that $\|y - z\| \leq \mu$ and, $\forall x \in \mathbb{R}^d$,*

$$f_{\gamma,\alpha}(x) \quad \leq \quad f(y) + \frac{L}{2\mu}\|x - z\|^2 + \frac{d}{2\alpha}\left(1 + \ln\left(1 + \frac{\alpha\mu L}{d}\right)\right), \tag{1}$$

*where $\gamma = \min\left\{\mu, \sqrt{\frac{\mu}{L\alpha}}\right\}$.*

In other words, for every local minima $y \in \mathbb{R}^d$ of the objective function $f$, $f_{\gamma,\alpha}(x)$ will be small (i.e., approximately $f(y)$) in a neighborhood of $y$ of size $\mu$ (see Fig. 1.c for an example). As a consequence, a good but thin local minimum will have its basin of attraction increased, and thus be easier to reach by GD even when the starting point $x_0$ is far from the local minimum. For example, consider $f(x) = \min\{1, |x|\}$. Its gradient is zero for any $x \notin [-1, 1]$, which means that GD initialized outside this region will be stationary. Moreover, for large smoothing parameters $\gamma$, RS will tend to flatten the objective and thus lead to the same behavior. However, BMR (with a sufficiently large $\alpha$) will create an almost quadratic function in a region $x \in [-\mu, \mu]$, thus allowing GD to converge even when initialized at distance $\mu$ from the origin.

**Experiments (REVIEWER 2, 3 AND 4)** *Adding the comparison against ME would have been helpful*: Moreau Envelope is a great tool when the inner optimization problem (*i.e.*, the proximal operator) can be solved efficiently, ideally in closed form. Here we focus on settings where there is no closed form, thus directly using Moreau Envelope would add a full optimization problem at each step. The results would eventually need to be analyzed w.r.t. the precision achieved when solving the ME problem, adding a layer of complexity to the analysis. We chose not to follow that direction as a first go, in order to illustrate the core concepts unequivocally.

*Optimization for DL, 10 dimensions is a bit disappointing*: Our goal was here to emphasize the improvements over RS on simple functions. Scaling the method to big ML problems such as the training of deep neural networks and policy optimization in Reinforcement Learning is currently investigating and left for future work.

*BMR presents some oscillations on Figure 3*: this is due to the exploratory nature of the method in addition to the particular choice of $\alpha$ in this experiment, leading to an opportunistic behavior. Lowering $\alpha$ would decrease the oscillations observed toward the end of the process.

**Typos and references (REVIEWER 1, 2 AND 3)** Thank you for pointing to the relevant works [1, 3]. We will include them in the paper. We will also fix the reference [2] in the proofs. Finally, the bias in $O(\sqrt{K})$ is indeed a typo, it will be modified to $O(1/\sqrt{K})$.

**General comments (REVIEWER 3)** Our method bridges the gap between two different smoothing methods commonly used in optimization, both having their advantages and drawbacks. It must not be viewed as a trivial combination of ME and RS but as an interpolation between them. This interpolation allows to take the best of both worlds and leads to faster optimization, as shown in the experiments.

# References

[1] A. Beck and M. Teboulle. Smoothing and first order methods: A unified framework. *SIAM Journal on Optimization*, 2012.

[2] C. Forbes, M. Evans, N. Hastings, and B. Peacock. *Statistical distributions*. John Wiley & Sons, 2011.

[3] Y. Nesterov. Smooth minimization of non-smooth functions. *Mathematical programming*, 2005.


[Meta-Review · NeurIPS 2020]

This paper gives an interesting new smoothing technique. The paper shows that the new smoothing can be much more accurate than randomized smoothing, and has an interesting effect of widening local minima. Overall the reviewers find the new technique very interesting and potentially useful in optimization.